Schistosome tegumental ecto-apyrase (SmATPDase1) degrades exogenous pro-inflammatory and pro-thrombotic nucleotides

Da’dara Akram A.
Bhardwaj Rita
Ali Yasser B.M.
Skelly Patrick J. Patrick.Skelly@Tufts.edu
Molecular Helminthology Laboratory, Department of Infectious Disease and Global Health, Cummings School of Veterinary Medicine, Tufts University , North Grafton, MA , USA
Jolly Christopher
Electronic publication date: 2014 Mar 18
Publication date: 2014
Volume: 2
Electronic Location ID: e316
Received 2014 Jan 13; Accepted 2014 Mar 4
Copyright: © 2014 Da’dara et al.
Copyright year: 2014
Copyright holder: Da’dara et al.
License: This is an open access article distributed under the terms of the Creative Commons Attribution License, which permits unrestricted use, distribution, and reproduction in any medium, provided the original author and source are credited.
License URL: https://creativecommons.org/licenses/by/3.0/

Keywords: Schistosoma, Trematode, Tegument, Ecto-enzyme, Apyrase, ATP, ADP, DAMP

Funding: NIH-NIAID AI-056273 This work was supported by grant AI-056273 from the NIH-NIAID and by a scholarship from the United States Agency for International Development/Egyptian Ministry of Higher Education (Mission Dept.). The funders had no role in study design, data collection and analysis, decision to publish, or preparation of the manuscript.

==============================
Schistosomes are parasitic worms that can survive in the hostile environment of the human bloodstream where they appear refractory to both immune elimination and thrombus formation. We hypothesize that parasite migration in the bloodstream can stress the vascular endothelium causing this tissue to release chemicals alerting responsive host cells to the stress. Such chemicals are called damage associated molecular patterns (DAMPs) and among the most potent is the proinflammatory mediator, adenosine triphosphate (ATP). Furthermore, the ATP derivative ADP is a pro-thrombotic molecule that acts as a strong activator of platelets. Schistosomes are reported to possess at their host interactive tegumental surface a series of enzymes that could, like their homologs in mammals, degrade extracellular ATP and ADP. These are alkaline phosphatase (SmAP), phosphodiesterase (SmNPP-5) and ATP diphosphohydrolase (SmATPDase1). In this work we employ RNAi to knock down expression of the genes encoding these enzymes in the intravascular life stages of the parasite. We then compare the abilities of these parasites to degrade exogenously added ATP and ADP. We find that only SmATPDase1-suppressed parasites are significantly impaired in their ability to degrade these nucleotides. Suppression of SmAP or SmNPP-5 does not appreciably affect the worms’ ability to catabolize ATP or ADP. These findings are confirmed by the functional characterization of the enzymatically active, full-length recombinant SmATPDase1 expressed in CHO-S cells. The enzyme is a true apyrase; SmATPDase1 degrades ATP and ADP in a cation dependent manner. Optimal activity is seen at alkaline pH. The Km of SmATPDase1 for ATP is 0.4 ± 0.02 mM and for ADP, 0.252 ± 0.02 mM. The results confirm the role of tegumental SmATPDase1 in the degradation of the exogenous pro-inflammatory and pro-thrombotic nucleotides ATP and ADP by live intravascular stages of the parasite. By degrading host inflammatory signals like ATP, and pro-thrombotic signals like ADP, these parasite enzymes may minimize host immune responses, inhibit blood coagulation and promote schistosome survival.

Introduction

Schistosomes are intravascular worms, commonly known as blood flukes, that cause the debilitating disease schistosomiasis. Over 200 million people are estimated to be infected with these worms globally and more than 600 million live at risk of infection (Vennervald & Dunne, 2004). Disease caused by Schistosoma mansoni is characterized clinically by abdominal pain, diarrhea, portal hypertension, anemia and chronic hepatic and intestinal fibrosis (Gryseels et al., 2006).

Mature male schistosomes are approximately 10 mm long and possess a ventral groove called the gynaecophoric canal in which the longer, cylindrical adult female often resides. In cross section, the male/female pair spans about 1 mm. Both sexes possess a pair of suckers (an anterior oral sucker and a ventral sucker) that are used for attachment to the blood vessel lining and to facilitate intravascular movement (Hockley & McLaren, 1973). Large tubercles are present on the dorsal surface of male S. mansoni, posterior to the ventral sucker. Tubercles are studded with prominent, rigid spines composed of actin bundles (Cohen et al., 1982). Female worms possess relatively few spines and their surface, while smoother and lacking large tubercles, is otherwise similar to the pitted and ridged surface of the male (Senft, Philpott & Pelofsky, 1961; Silk, Spence & Gear, 1969).

S. mansoni adult worms wander extensively within the complex venous system draining the intestinal tract (Pellegrino & Coelho, 1978). Both single and paired worms move constantly along the vessels (Bloch, 1980). The relatively large adults enter blood vessels whose diameter is equivalent to their own (Bloch, 1980). In addition, the worms can elongate considerably to enter even smaller vessels, such as the mesenteric venules, to lay eggs (Bloch, 1980).

Parasite suckers, tubercles and spines used for migration in the bloodstream can impinge on host vascular endothelia (Smith & von Lichtenberg, 1974). In addition the large, mature schistosomes moving through small blood vessels hamper and alter blood flow (Bloch, 1980), almost certainly causing sheer stress and restricting local O2 concentration. All of these conditions, leading to endothelial cell stress, may trigger the release by these cells of endogenous distress signals. These signals, known collectively as damage-associated molecular patterns (DAMPs), indicate tissue damage to the host and can initiate primary immune responses. Extracellular nucleotides such as ATP are known to function as potent DAMPs by acting as endogenous tissue-derived signaling molecules that contribute to inflammation and immunity. Following tissue damage or during inflammation, or when exposed to shear stress, many cells release ATP (Hanley et al., 2004; Lohman, Billaud & Isakson, 2012). There is a substantial literature demonstrating that extracellular ATP can function as a proinflammatory immunomediator by acting on multiple immunological effector cell types including neutrophils, macrophages, dendritic cells, and lymphocytes (Reviewed in Bours et al., 2006; Hanley et al., 2004; Yegutkin, 2008).

General activation of the immune system following exposure to DAMPs can be controlled by their degradation in a timely manner. For instance, concentrations of ATP in the extracellular compartments of vertebrates are regulated by the following membrane-bound, nucleotide-metabolizing ecto-enzymes: alkaline phosphatase, phosphodiesterase and ATP-diphosphohydrolase (Bours et al., 2006; Burnstock, 2006). ATP degradation in this manner helps prevent uncontrolled inflammation and averts collateral cell damage.

As noted, schistosomes in the vasculature may directly and indirectly stress the endothelium which could lead to the release of the DAMP, ATP (Bhardwaj & Skelly, 2009). This would then stimulate inflammatory immune responses in the vicinity of the worms that could debilitate and kill them. However, it has been shown that schistosomes, like their hosts, express a panel of ecto-enzymes that could catabolize ATP. These are alkaline phosphatase (SmAP), phosphodiesterase (SmNPP-5) and ATP-diphosphohydrolase (SmATPDase1) (Bhardwaj & Skelly, 2009). We hypothesize that these parasite tegumental enzymes specifically counteract ATP DAMP-mediated inflammatory signaling and limit the host’s attempts to focus inflammatory mediators around the worms (Bhardwaj & Skelly, 2009). In this manner, these tegumental molecules help impair host immune defenses and promote parasite survival.

In addition to contending with host immunity, intravascular schistosomes, which act as obstructions in the blood vessels, also need adaptations to avoid promoting blood coagulation in their vicinity. The ectoenzymes under study here may exert a key regulatory influence on these processes too. Platelets play a central role in blood clotting and ATP can regulate platelet reactivity by way of direct action on platelet purinergic receptors (Mahaut-Smith et al., 2000). In addition, the first step in ATP hydrolysis leads to the generation of ADP and ADP is a major agonist of platelet recruitment and aggregation (Gachet, 2006). Furthermore, platelets themselves can damage schistosomes (Joseph et al., 1983). Therefore the catabolism of ATP and ADP via SmAP, SmNPP-5 and/or SmATPDase1 may additionally lead to the inhibition of platelet aggregation and thrombus formation around the worms.

It has long been known that schistosome tegumental extracts do possess ATP and ADP hydrolyzing activity (Vasconcelos et al., 1993). Electron microscopy analysis identified electron-dense lead phosphate deposits on the outer surface of adult parasites upon hydrolysis of ATP or ADP and the production of inorganic phosphate (Vasconcelos et al., 1996; Vasconcelos et al., 1993). These data suggest that the activity is external to the body of the worm but do not identify the enzyme(s) responsible. One candidate is schistosome alkaline phosphatase (SmAP). The cDNA encoding SmAP was recently cloned and characterized (Bhardwaj & Skelly, 2011). SmAP is a ∼62 kDa glycosylphosphatidylinositol (GPI) anchored protein that is expressed in the tegument and internal tissues of the adult worms (Bhardwaj & Skelly, 2011; Cesari, 1974; Dusanic, 1959; Levi-Schaffer et al., 1984; Morris & Threadgold, 1968; Pujol et al., 1990). The protein can be cleaved from cultured schistosomula (Espinoza et al., 1988) and from adult worms (Castro-Borges et al., 2011) by the phosphatidylinositol-cleaving enzyme—phosphatidylinositol-specific phospholipase C. Tegumental proteomic analysis confirms that SmAP is found in the schistosome surface membranes (Braschi et al., 2006; van Balkom et al., 2005) and is available for surface biotinylation (Braschi & Wilson, 2006).

Proteomic analysis of tegument preparations revealed a second potential ATP and ADP hydrolyzing enzyme there, specifically a phosphodiesterase designated SmNPP-5 that could also be biotinylated at the adult parasite surface (Braschi et al., 2006; Braschi & Wilson, 2006). SmNPP-5 is a ∼53 kDa protein possessing a single C-terminal transmembrane domain that is expressed exclusively in the intra-mammalian life stages (Rofatto et al., 2009). The protein is expressed highly in the adult tegument and exhibits a unique clustered localization pattern in the tegument as revealed by immunoEM analysis (Bhardwaj et al., 2011).

A third candidate tegumental, ATP- and ADP-cleaving enzyme is the ATP diphosphohydrolase homolog SmATPDase1 (Vasconcelos et al., 1996; Vasconcelos et al., 1993). This ∼63 kDa protein possesses an N-terminal and a C-terminal transmembrane domain. It was detected in the adult tegument by immunolocalization (DeMarco et al., 2003; Levano-Garcia et al., 2007) and was identified in adult tegument extracts by proteomic analysis (Braschi et al., 2006; van Balkom et al., 2005). Like SmAP and SmNPP-5, SmATPDase1 was also available for surface biotinylation (Braschi & Wilson, 2006).

In this work we set out to determine whether degradation of the proinflammatory DAMP, ATP, as well as its pro-thrombotic derivative ADP could be mediated by any, or all, of these schistosome enzymes (SmAP, SmNPP-5 and SmATPDase1). We aimed to determine if schistosomes, like their hosts, exhibit redundancy with regard to exogenous ATP and ADP breakdown. In this work we employed RNAi to suppress the expression of the genes encoding these enzymes in order to measure the ability of each to cleave ATP and ADP.

Materials and Methods

Parasites

Snails were provided by the Schistosome Research Reagent Resource Center for distribution by BEI Resources, NIAID, NIH: Schistosoma mansoni, strain NMRI exposed Biomphalaria glabrata snails, strain NMRI, NR-21962. Cercariae were obtained from infected B. glabrata and isolated parasite bodies were prepared as described (Skelly, Da’dara & Harn, 2003). Parasites were cultured in complete DMEM/F12 medium supplemented with 10% heat-inactivated fetal bovine serum, 200 U/ml penicillin and 200 µg/ml streptomycin, 0.2 µM Triiodo-L-thyronine, 1.0 µM serotonin and 8 µg/ml human insulin. Parasites were maintained at 37 °C, in an atmosphere of 5% CO2. Adult male and female parasites were recovered by perfusion from Swiss Webster mice that were infected with 125 cercariae, 7 weeks previously. Work with animals was approved by the Tufts University IACUC; protocol number: G2012-150.

Treatment of parasites with siRNAs

Schistosomula and adult worms were treated with synthetic siRNAs targeting SmAP (GenBank accession number EU040139), SmNPP-5 (GenBank accession number EU769293) and SmATPDase1 (GenBank accession number AY323529). An “irrelevant siRNA” was used as a control and its sequence has no identity in the S. mansoni genome. The siRNAs were obtained from IDT, Coralville, IA. The siRNAs targeting SmAP, SmNPP-5, and SmATPDase1 are the following: SmAP: 5′-AAGAAATCAGCAGATGAGAGATTTAAT-3′, SmNPP-5: 5′-TTGATGGATTTCGTTATGATTACTTTG-3′, SmATPDase1: 5′- GGACUUUAUGGUUGGGUAUCAGUGA-3′. The control, irrelevant siRNA is: 5′-CT TCCTCTCTTTCTCTCCCTTGTGA-3′.

To deliver the siRNAs, parasites (1000 schistosomula or 10–12 adults/group) in 50–100 µl electroporation buffer (BioRad, CA) containing 2.5–10 µg siRNA, were electroporated in a 4 mm cuvette by applying a square wave with a single 20-ms impulse, at 125 V and at room temperature, as described (Krautz-Peterson et al., 2007; Ndegwa, Krautz-Peterson & Skelly, 2007). To suppress SmAP, SmNPP-5 and SmATPDase1 together, 5 µg of SmAP and SmNPP-5 siRNA and 10 µg SmATPDase1 siRNA were used in the case of adults; 2.5 µg of each siRNA was used in the case of schistosomula. In these experiments, an equivalent amount of the irrelevant siRNA was used in the control group. Parasites were transferred to 500–1300 µl complete DMEM/F12 medium after electroporation. After overnight culture, medium was replaced with fresh rich medium (complete DMEM/F12).

Gene expression analysis

To assess the level of target gene suppression post-siRNA treatment, RNA and protein were isolated from worm lysates using the PARIS kit (Applied Biosystems, CA). Samples were homogenized on ice using an RNase free pestle for ∼1 min and the parasite homogenates were split into two halves. One half was used to isolate RNA and the other for protein analysis. RNA was isolated from the parasite homogenate using the PARIS Kit, as per the manufacturer’s guidelines. Residual DNA was removed by DNase digestion using a TurboDNA-free kit (Applied Biosystems, TX). cDNA was synthesized using 1 µg RNA, an oligo (dT)20 primer and Superscript III RT (Invitrogen, CA). Gene expression of SmAP, SmNPP-5 and SmATPDase1 was measured by quantitative real time PCR (qRT-PCR), using custom TaqMan gene expression systems from Applied Biosystems, CA. The primers and probes employed in this research are listed in Table 1. The procedure, involving total RNA extraction and quantitative real time PCR, has been described (Krautz-Peterson et al., 2007; Ndegwa, Krautz-Peterson & Skelly, 2007). Alpha tubulin was used as the endogenous control gene for relative quantification, as described (Krautz-Peterson et al., 2010), employing the ΔΔCt method (Livak & Schmittgen, 2001). Results obtained from parasites treated with irrelevant siRNA were used for calibration. For graphical representation, the ΔΔCt values were normalized to controls and expressed as a percentage difference.

Table 1 Sequences of oligonucleotides used in qRT-PCR analysis.

For each gene a forward (F) and reverse (R) primer were used in conjunction with a FAM dye labeled probe.

Gene	Primer name	Sequence	
SmAP	SmAP-F	5′-GCCATCCGACAAGGAATATAAGTGT-3′	
Sm-AP-R	5′-GGTCCATTGAAAAAGGAGGATATGAGA-3′	
Sm-AP-FAM	5′-FAM-ATCTCCTTTTGCAGTATTATC-3′	
SmNPP-5	SmNPP-5-F	5′-GGACGATTATTGCTGACAGAACGT-3′	
	SmNPP-5-R	5′-TGGAGACATCTCTTTGTAATCTGGATCA-3′	
	SmNPP-FAM	5′-FAM-TTTATTTTTCAGGGTTATCCC-3′	
SmATPDase1	SmATPd-F	5′-CTGATGCCGTTATGAAGTTTTGCA-3′	
	SmATPd-R	5′-ACCTTCAGCAAGTGCATGTTGA-3′	
	SmATPd-FAM	5′-FAM-AAAGATGTGGCTAAAATT-3′	
α-Tubulin	Tub-F	5′-GGTTGACAACGAGGCCATTTATG-3′	
	Tub-R	5′-GCAGTAAACCCTTGGTCAGATAATTTTG-3′	
	Tub-Probe	5′-FAM-ATATTTGTCGACGGAAT-3′	

Anti-SmAP, anti-SmNPP-5 and anti-SmATPDase1 antibody production

Anti-SmAP and anti-SmNPP-5 antibodies were generated in rabbits using, in each case, a synthetic peptide as immunogen (Bhardwaj et al., 2011; Bhardwaj & Skelly, 2011). Anti-SmATPDase1 antibody, generated in mice against recombinant SmATPDase1 protein, was a kind gift from Dr. Sergio Verjovski-Almeida, University of Sao Paulo, Brazil (DeMarco et al., 2003).

Western blotting analysis

To monitor protein levels, parasite samples were first homogenized on ice in ice-cold cell disruption buffer (PARIS Kit) followed by incubation for 30 min on ice to yield total parasite lysate. Protein content was measured using the BCA Protein Assay Kit (Pierce, IL) according to the manufacturer’s instructions. Soluble protein (5 µg in 20 µl SDS-PAGE sample buffer) was subjected to SDS-PAGE under reducing conditions, blotted onto PVDF membrane and blocked using 5% skim milk in PBS containing 0.1% Tween 20 (PBST) for 1 h at room temperature. The membrane was then probed overnight at 4 °C with anti-SmAP (1:400), or anti-SmNPP-5 (1:200) or SmATPDase1 antiserum (1:10). Following 3 washes with PBST and incubation with donkey anti-rabbit IgG conjugated to horse radish peroxidase (HRP) (GE Healthcare, UK), diluted 1:5000, (for SmAP, and SmNPP-5) and goat anti-mouse IgG conjugated to HRP (Invitrogen) diluted 1:2000, (for SmATPDase1) for 1 h at 37 °C. Protein bands were visualized using ECL Western Blotting Detection Reagents (GE Healthcare) and X-ray film (ISC BioExpress, Belgium). The same membrane was probed three times to detect SmAP, SmNPP-5 and SmATPDase1. For each re-use, the bound antibody was striped using Restore Western Blot Stripping Buffer from Thermo Scientific (IL, USA) for 4 h at 37 °C and then washed in PBS twice for 30 min each. To monitor protein loading per lane, a duplicate gel was stained with Coomassie Brilliant Blue, to visually ensure roughly equivalent protein loading per sample.

Cloning and transient expression of SmATPDase1 in CHO-S cells

The complete coding region of SmATPDase1 (accession number AY323529) was codon optimized for expression in hamster and mouse cells by Genscript and cloned into pUC57 (Genscript USA Inc., Piscataway, NJ). Using this DNA, two constructs were generated for protein expression in mammalian cells: (1) the full-length open reading frame (illustrated in Fig. 4A) was excised from the pUC57 plasmid using the restriction enzymes NheI and XhoI. These enzyme sites were introduced into the sequence during gene synthesis. The excised DNA was then cloned into the pSecTag2A expression plasmid (Invitrogen) that had been previously digested with the same restriction enzymes. (2) The region encoding just the large extracellular region of SmATPDase1 (encompassing amino-acids S66–Q507 and lacking both transmembrane domains, indicated in Fig. 4A) was amplified by PCR using AccuPrime High Fidelity Taq DNA polymerase (Invitrogen) and cloned at the AscI and XhoI sites in frame with the Igk-leader sequence in the pSecTag2A expression plasmid. All cloned DNAs were sequenced to verify successful in-frame cloning.

CHO-S cells grown in suspension (Invitrogen) were used for transient SmATPDase1 protein expression. The cells were grown in 30 ml of serum-free Free-Style CHO-S expression medium to 1 × 106 cells/ml. Cells were then transfected with 1 µg plasmid DNA/ml using Free-Style MAX Transfecting agent according to the manufacturer’s instructions (Invitrogen). Seventy two hours later, cells were harvested by centrifugation and cells and culture supernatants were analyzed for SmATPDase1 protein expression. Cell lysates were prepared by cell sonication (3 times, 30 s each) on ice in assay buffer (20 mM HEPES buffer, pH 7.4, 1% Triton X-100, 0.135 M NaCl, 5 mM KCl, 1 mM CaCl2). Lysates were incubated on ice for 1 h, centrifuged at 4 °C for 20 min at maximum speed. Protein concentration in the recovered supernatants was determined using a BCA kit (Pierce).

SmATPDase1 assay

Both ATPase and ADPase activities of the recombinant protein were assayed in 96-well microtiter plates at 37 °C for 30–120 min. The standard 200 µl assay buffer contains 20 mM HEPES buffer, pH 7.4, 1% Triton X-100, 0.135 M NaCl, 5 mM KCl, 1 mM CaCl2, and recombinant SmATPDase1. Reactions were initiated by the addition of ATP or ADP nucleotide solution to a final concentration of 2 mM. At different time points thereafter, 10 µl aliquots were transferred to 190 µl ice-cold water, and stored at −20 °C until analyzed. The amount of inorganic phosphate (Pi) released by the enzyme was determined using a Phosphate Colorimetric Assay Kit (BioVision) according to the manufacturer’s instructions. Activity was calculated by subtracting the minimal, nonspecific ATP or ADP hydrolysis that was detected in the absence of the enzyme. Nucleotide hydrolysis was linear with time under the assay conditions used and was proportional to the amount of enzyme used. The linear amount of the enzyme was always determined by performing preliminary assays with different amounts of cell lysate (containing 5–50 µg protein). An equivalent amount of lysate from control or mock transfected cells served as control.

SmATPDase1 assays using live parasites and CHO-S cells

ATP and ADP hydrolysis activities of live parasites, or intact CHO-S cells (expressing recombinant SmATPDase1), or mock transfected cells, were determined as described above with slight modifications. Briefly, live parasites, or CHO-S cells, were first washed 3 times in isotonic wash solution (20 mM HEPES buffer, pH 7.4 containing 0.13 M NaCl, 5 mM KCl, 1 mM CaCl2, 10 mM Glucose). Next, a specific number of parasites or cells were resuspended in 100 µl isotonic wash solution. Reactions were started by the addition of a 100 µl of the same buffer containing ATP or ADP to produce a final concentration of 2 mM. Released inorganic phosphates were measured using the Phosphate Colorimetric Assay Kit (BioVision) according to the manufacturer’s instructions.

Characterization of recombinant SmATPDase1 (rSmATPDase1)

Enzyme (10 µg rSmATPDase1) activity was measured in buffer containing 20 mM HEPES, pH 7.4, 1% Triton X-100, 0.135 M NaCl and 5 mM KCl. In some cases this buffer was supplemented with either 1 mM CaCl2 or 1 mM MgCl2 or 1 mM EDTA plus 1 mM EGTA or 1 mM CaCl2 plus 10, 50 or 100 µM thapsigargin. Reaction conditions were as described above (SmATPDase1 assay).

Km values for rSmATPDase1 were determined in the standard assay buffer (described earlier) containing different substrate concentrations (0–2.5 mM) of ATP or ADP. Km values were calculated using computerized nonlinear regression analysis of the data fitted to the Michaelis–Menten equation using Graphpad Prism 4.0.

The effect of pH on ATP and ADP hydrolysis by rSmATPDase1 was determined in a 200 µl enzyme assay using a wide-range buffer system covering the pH range of 5.5–10.0 (MES, pH 5.5–6.5; MOPS, pH 6.5–7.5; HEPES, pH 7.0–8.0, Tris-HCl, pH 7.5–9.0; Trizma, pH 9.0; Glycine-NaOH, pH 9.0–10). Assay solutions contained 20 mM buffer, 1% Triton X-100, 0.135 M NaCl, 5 mM KCl, 1 mM CaCl2, and 2 mM ATP or ADP with 10 µg cell lysate. The reaction was carried out for 30–120 min. Aliquots containing released Pi were assayed at different time points using the phosphate colorimetric assay, as above.

Data analysis

For qRT-PCR and Pi release assay data, one way analysis of variance (ANOVA) and Tukey as the post hoc test was used. Other data were analyzed using the Student’s t-test. In all cases, differences were considered significant when P values <0.05.

Results

Cleavage of exogenous nucleotides by schistosomes

Living schistosomes possess the ability to catabolize exogenous nucleotides. When live adult males are incubated in the presence of ATP, ADP or AMP they cleave these molecules resulting in the release of inorganic phosphate (Pi), as shown in Fig. 1A. Likewise, groups of living schistosomula incubated with ATP, ADP or AMP cleave these nucleotides (Fig. 1B). In the case of both adults and schistosomula, most Pi is generated with ATP as substrate, least is generated with AMP as a substrate and an intermediate amount from ADP. Under the conditions used, we detect no background generation of inorganic phosphate in control samples lacking parasites when ATP and AMP are used and only negligible levels of Pi (<5 nmol) when ADP is added. One model for schistosome catabolism of these metabolites suggests that three different enzymes with overlapping function may be involved. This proposed pathway for catabolism of exogenous ATP, ADP and AMP by intravascular schistosomes is shown in Fig. 1C.

SmAP, SmNPP-5 and SmATPDase1 gene suppression using RNAi

In order to uncover which of the enzymes is involved in each step of the exogenous nucleotide catabolism pathway shown in Fig. 1C, the genes encoding these enzymes were first subjected to suppression using RNAi. Suppression was monitored by qRT-PCR 7 days after treatment and results are shown in Figs. 2A–2C. In each case, gene expression is depicted relative to the control group treated with an irrelevant siRNA (set at 100%, grey bars in Fig. 2). Relative to the control, it is clear that all 3 targeted genes have been well suppressed (P < 0.05, in each case). Figure 2A illustrates results for SmAP; the group treated with a specific SmAP siRNA exhibits ∼90% lower SmAP gene expression relative to the control group. This is the case when the SmAP siRNA is used alone (lane SmAP, Fig. 2) or in combination with siRNAs also targeting SmNPP-5 and SmATPDase1 (lane marked “All 3”, Fig. 2A). Similarly, from Fig. 2B it is clear that when SmNPP-5 is targeted with specific SmNPP-5 siRNA, >90% suppression is observed. Again, this is the case both when SmNPP-5 is targeted with SmNPP-5 siRNA alone (lane SmNPP-5, Fig. 2B) or with siRNAs also targeting SmAP and SmATPDase1 (lane “All 3”, Fig. 2B). Finally, similar results are seen for SmATPDase1 in Fig. 2C; in this case ∼80% suppression is seen when this gene is targeted with SmATPDase1 siRNA either alone (lane SmATPD, Fig. 2C) or in addition to siRNAs targeting SmAP and SmNPP-5 (“All 3”, Fig. 2C). Gene knockdown was specific; siRNAs targeting SmAP have no significant effect on SmNPP-5 or SmATPDase1 levels compared to control; suppressing SmNPP-5 did not appreciably impact the SmAP or SmATPDase1 genes. In a similar manner, targeting SmATPDase1 led to its specific knockdown without significant impact on the SmAP or SmNPP-5 genes. Suppression was consistently better for SmAP and SmNPP-5 (>90%) versus SmATPDase1 (∼80%). Our attempts to suppress the SmATPDase1 gene still further by using greater amounts of siRNA (up to 35 µg), or different siRNAs, were not successful (data not shown).

Figure 1 Ecto-nucleotidase activity in schistosomes.

(A) Phosphate (nmol) release (mean ± SE, n = 8) following the addition of ATP, ADP or AMP (2 mM) to individual live adult male worms over 3 h. (B) Phosphate (nmol) release (mean ± SE, n = 3) following the addition of ATP, ADP or AMP (2 mM) to 1,000 schistosomula over 1 h. In the absence of parasites, no or negligible (<5 nmol) Pi is detected. The data shown are representative of at least three independent experiments. (C) The proposed pathway in schistosomes for exogenous ATP catabolism via ADP and AMP to adenosine. The following three schistosome tegumental ectoenzymes are hypothesized to be involved: SmAP (S. mansoni alkaline phosphatase), SmNPP-5 (S. mansoni nucleotide pyrophosphatase-phosphosdiesterase-5) and SmATPDase1 (S. mansoni ATP diphosphohydrolase1).

Figure 2 Suppression of schistosome ectoenzyme genes using RNAi.

Relative SmAP (A), SmNPP-5 (B) and SmATPDase1 (C) gene expression (mean ± SD, n = 3) in schistosomula treated with SmAP, SmNPP-5, SmATPDase1 (SmATPD), control (grey bar) or no (None) siRNA. One group was treated with siRNAs simultaneously targeting the three ectoenzyme genes (SmAP and SmNPP-5 and SmATPDase1, lane marked “All 3”). In all cases target gene suppression is significantly different from control (P < 0.05). (D) Western blotting analysis in which protein extracts of parasites treated either with siRNAs simultaneously targeting the three ectoenzyme genes (SmAP and SmNPP-5 and SmATPDase1, lane marked “All 3”) or control siRNA or no siRNA (None) are probed with antibody specific for SmAP (top panel), or SmNPP-5 (second panel), or SmATPDase1 (third panel). The bottom panel shows a fragment of the gel stained with Coomassie blue to ensure roughly equal protein loading per lane. The data shown are representative of four independent experiments.

In order to assess the impact of gene suppression at the protein level, target-specific antibodies were used in western blotting analyses and results are shown in Fig. 2D. Protein extracts of control and the triply-suppressed parasites (SmAP, SmNPP-5 and SmATPDase, lane “All 3” in Fig. 2D) were probed with anti-SmAP, anti-SmNPP-5 or anti-SmATPDase1 antibodies. It is clear that, in all cases, the siRNA treatment resulted in a diminution in protein levels compared to parasites treated either with irrelevant, control siRNA (Fig. 2D, control) or with no siRNA (Fig. 2D, None). This is the case for SmAP (Fig. 2D, top row), SmNPP-5 (second row) and SmATPDase1 (third row). The bottom panel in Fig. 2D shows a fragment of a Coomassie Blue stained polyacrylamide gel, distant from the location of any of the targets, to illustrate that all lanes contained roughly equivalent amounts of parasite protein.

Parasites with each of the surface enzyme genes suppressed (either separately or all together) exhibited no morphological differences compared to controls. This suggests that high levels of expression of these genes are not very important for the worms in culture.

SmATPDase1 alone is responsible for exogenous ATP and ADP degradation

The ability of suppressed or control parasites in culture to degrade exogenously added ATP (2 mM) was measured over time. The rate of Pi release per parasite in culture is shown in Fig. 3A. Each control parasite treated with an irrelevant siRNA generates an average of ∼67 nmol Pi/h (Fig. 3A, Control). Likewise, control parasites treated with no siRNA (Fig. 3A, None) as well as parasites whose SmAP gene or SmNPP-5 gene has been suppressed (Fig. 3A, SmAP and SmNPP-5) all generate a similar amount of Pi. In contrast, parasites whose SmATPDase1 gene has been suppressed (Fig. 3A, SmATPD, grey bar) are significantly impaired in their ability to cleave exogenous ATP and liberate Pi (P < 0.05); only about 50% of the ATPase activity was detected, compared to controls. These data show that SmATPDase, but not SmAP or SmNPP-5, degrades exogenous ATP.

Figure 3 Apyrase activity of ecto-enzyme suppressed and control parasites.

Enzyme activity (phosphate (Pi) release per hour, mean ± SE, n = 8) from individual, living adult male schistosomes treated with the indicated siRNAs and incubated with 2 mM ATP (A) or ADP (B). In both cases, significantly lower activity is seen in parasites treated with siRNA targeting SmATPDase1 (grey bars) compared to all other groups (P < 0.05). The data shown are representative of three independent experiments.

Next, the ability of suppressed or control parasites in culture to degrade exogenously added ADP (2 mM) was measured over time. Again, the amount of Pi released in culture was measured and results are shown in Fig. 3B. The data for Pi release, when ADP is the substrate, are broadly similar to those obtained when ATP is used. Parasites whose SmATPDase1 gene is suppressed (Fig. 3B, SmATPD, grey bar) again generate about 50% of the Pi released by those parasites treated with a control, irrelevant siRNA (Fig. 3B, control, P < 0.05). Parasites whose SmNPP-5 gene or SmAP gene were suppressed (Fig. 3B, SmNPP-5 and SmAP) generate Pi at a rate not significantly different from the control treated group. These data show that, as for ATP, SmATPDase1, not SmAP or SmNPP-5, degrades exogenous ADP.

Characterization of recombinant SmATPDase1 expressed in CHO cells

From the data presented, it is clear that SmATPDase1 is a key enzyme in the catabolic pathway under study. In order to characterize the enzyme further, efforts were made to express the protein in CHO-S cells in two different forms—in full length form (from residue 1 through 544, as illustrated in Fig. 4A, top panel) and as a secreted form lacking the predicted N-terminal and C-terminal transmembrane (TM) domains (i.e., from residue S66 through Q507, indicated in Fig. 4A, bottom panel). Roughly 72 h after cell transfection with plasmid constructs expressing the full-length or the secreted form, ATP or ADP was added to CHO cell lysates (containing 10 µg protein) and Pi release measured over time. Control cells were not transfected with any plasmid. As shown in Figs. 4B and 4C, only lysate from cells expressing the full length protein exhibited activity. This was the case following either ATP addition to the assay (Fig. 4B, grey bar) or following ADP addition to the assay (Fig. 4C, grey bar). Any secreted protein was inactive; lysate from cells targeted to express the secreted protein displayed activity indistinguishable from that of control cell lysate (black versus white bars, Figs. 4B and 4C).

Figure 4 Expression of recombinant SmATPDase1.

(A) Depiction of the full length 544 amino acid SmATPDase1 protein (top) which contains two transmembrane (TM) domains. Numbers refer to amino acid residues. A truncated version of the protein from residues S66 to Q507, lacking both TM domains, and predicted to be secreted following expression in CHO cells is depicted below. (B) ATPase activity (mean ± SE, n = 3) in CHO-S cell lysates (10 µg/assay) three days after transfection with a full length or secreted or no (None) DNA construct. The inset shows ATPase activity on the surface of living CHO-S cells (75 ×103 or 150 ×103) three days after transfection with a full length or no (control) DNA construct. (C) ADPase activity (mean ± SE, n = 3) in CHO-S cell lysates (10 µg/assay) three days after transfection with a full length or secreted or no (None) DNA construct. The inset shows ADPase activity on the surface of living CHO-S cells (75 ×103 or 150 ×103) three days after transfection with a full length or no (control) DNA construct. In all cases only the activity of the full length construct differs significantly from other groups (P < 0.05). The data shown in (B) and (C) are representative of five independent experiments.

The expectation is that some of the full length, recombinant SmATPDase1 (rSmATPDase1) ecto-enzyme will be expressed on the plasma membrane of the transfected CHO cells. To look for activity at the surface of living CHO cells, transfected and control cells were plated at 75 ×103 or 150 ×103 per well and either ATP (inset Fig. 4B) or ADP (inset, Fig. 4C) was added in a Pi release assay. It is clear that the living transfected cells (grey bars, Figs. 4B and 4C, insets) can cleave both ATP (Fig. 4B, inset) and ADP (Fig. 4C, inset) to release Pi at substantially greater levels than controls (white bars, Figs. 4B and 4C, insets). As expected, in both cases, greater numbers of cells used in the assay (150 ×103 versus 75 ×103, Figs. 4B and 4C, insets) yield proportionally greater Pi release.

The activity of rSmATPDase1 was measured under different experimental conditions. As demonstrated in Fig. 5, the catalytic activity exhibited by SmATPDase1 in the CHO cell lysate towards ATP (Fig. 5A) and ADP (Fig. 5D) was markedly increased by the addition of 1 mM Mg++ to the mixture and (for ATP) increased still further by the addition of 1 mM Ca++. Chelating these ions from the original lysate with the addition of EDTA and EGTA greatly reduced the activities detected (Figs. 5A and 5D). Adding thapsigargin to the lysate (at 10, 50 or 100 µM) had a minor inhibitory effect (∼20% at all concentrations tested) on ATP hydrolysis and an even smaller effect (∼5%) on ADP hydrolysis. Only results obtained using the highest thapsigargin concentration tested (100 µM) are shown. The Km of recombinant SmATPDase1 is 0.4 ± 0.02 mM for ATP (Fig. 5B) and 0.252 ± 0.02 mM for ADP (Fig. 5E). Both ATP and ADP catalytic activities display pH optima in the alkaline range; the ATPase activity is maximal at pH ≥ 8.5 (Fig. 5C) and the ADPase activity is maximal at pH ≥ 7.5 (Fig. 5F).

Figure 5 Characterization of recombinant SmATPDase1.

The top panel (A–C) deals with ATP and the lower panel (D–F) deals with ADP. ATPase activity (A) and ADPase activity (D) in cell lysates (mean ± SE, n = 3) expressing SmATPDase1 (10 µg protein) in the presence of added calcium (Ca++) or magnesium (Mg++) or nothing (None) or EDTA plus EGTA or Ca++ plus thapsigargin (Thaps, 100 µM). Michaelis-Menton plot of ATPase activity (B) and ADPase activity (E) in cell lysates expressing rSmATPDase1. The Km for ATP is 0.4 ± 0.02 mM and the Km for ADP is 0.252 ± 0.02 mM. The effect of pH on ATPase activity (C) and ADPase activity (F) in cell lysates expressing SmATPDase1. Data shown are representative of independent experiments performed at least 3 times.

Discussion

The migration of intravascular schistosomes can stress blood vessel endothelia (Bloch, 1980; Smith & von Lichtenberg, 1974) likely leading to the release of host molecules, such as ATP, that signal cell damage (Bhardwaj & Skelly, 2009). In the extracellular environment, ATP is a potent proinflammatory mediator and its byproduct (ADP) is potently pro-thrombotic. It has been hypothesized that schistosomes have evolved to impede host immunity and thrombus formation by degrading these host signaling molecules using nucleotide metabolizing enzymes expressed on their surface (Bhardwaj & Skelly, 2009).

It has long been known that schistosome tegumental extracts do possess ATP and ADP hydrolyzing capabilities and that living worms can deplete exogenous ATP and ADP (Vasconcelos et al., 1993). Here we confirm that living parasites (both adults and schistosomula) can degrade exogenous ATP, ADP and AMP. In the case of vertebrates, ectoenzymes belonging to three different classes are known to engage in the extracellular ATP degradation pathway (Bours et al., 2006). These are alkaline phosphatase, phosphodiesterase and ATPdiphosphohydrolase. In what appears to be considerable redundancy in vertebrates, enzymes belonging to these three classes can all mediate ATP and ADP breakdown while two of the three can mediate AMP breakdown (Bours et al., 2006). Using this literature as a guide, we hypothesized that the exogenous ATP degradation pathway in schistosomes could similarly be mediated by several known tegumental enzymes belonging to these enzyme classes. These are SmAP, SmNPP-5 and SmATPDase1.

In this paper, the hypothesis that schistosomes possess equivalent redundancy to vertebrates in their ability to degrade extracellular ATP and ADP was tested. First, RNAi was employed to suppress the expression of all 3 ectoenzyme genes (either alone or in combination). The expression of each gene is robustly and specifically suppressed both when that gene is targeted by itself or with other genes. Suppression at the RNA level is confirmed by quantitative real-time PCR analysis; suppression at the protein level is seen by western blotting analysis.

When parasites that have had all three ectoenzyme genes suppressed were maintained in culture for up to 4 weeks they exhibited no morphological differences when compared to controls. This suggests that normal expression of the genes encoding the three ectoenzymes is not essential for worm survival in culture and is in agreement with the hypothesis that these enzymes are primarily important for parasites within the vertebrate host where they act to minimize host purinergic signaling.

The first step in the pathway under study here is the catabolism of ATP to ADP. In order to decipher which of the three enzymes participate in this step, their genes were suppressed using RNAi. Next, the ability of the SmAP- or SmNPP-5- or SmATPDase1-suppressed parasites to degrade ATP (added to the assay buffer) was compared with the ability of controls to degrade ATP. The results are clear—of these 3 gene knockdown conditions, it is only following SmATPDase1 knockdown that parasites exhibit a reduced ability to cleave exogenous ATP, in comparison to controls. The SmAP- and SmNPP-5-suppressed adult parasites were not impacted in their ATP-hydrolyzing ability, which was comparable to the controls. Only the ATP degrading ability of the SmATPDase1-suppressed parasites was significantly reduced. Thus, unlike vertebrates, schistosomes utilize just one ectoenzyme to cleave ATP. There is no redundancy in schistosomes at this step.

The second step in the pathway involves the cleavage of ADP. A similar experiment to that just described for ATP was undertaken; the ability of SmAP- or SmNPP-5- or SmATPDase1-suppressed parasites versus controls to degrade ADP (added to the assay buffer) was compared. It was observed that the SmATPDase1-suppressed group alone exhibited a lessened ability to cleave ADP. The SmAP- and SmNPP-5 suppressed parasites had no impairment in ADP cleavage compared to controls. As for ATP cleavage, this second ADP-cleavage step is also non-redundant in schistosomes. SmATPDase1, in addition to being an ATPase, is also an ADPase.

The final step in the pathway is the cleavage of AMP to generate adenosine. In previous work it has been established that SmAP fulfills this function (Bhardwaj & Skelly, 2011). Thus the final pathway in schistosomes is simpler than that observed in vertebrates and is illustrated in Fig. 6. The third enzyme, SmNPP-5 does not participate in this pathway. While its function at the surface of the intravascular worms is not known, SmNPP-5 does fulfill an important role for schistosomes since it has been shown that parasites whose SmNPP-5 gene is suppressed fail to establish a robust infection in mice (Bhardwaj et al., 2011).

Figure 6 The pathway in schistosomes for exogenous ATP catabolism via ADP and AMP to adenosine.

Work reported here demonstrates that, of the three ectoenzyme candidates, only SmATPDase1 (S. mansoni ATP diphosphohydrolase1) can cleave ATP and ADP. In the final step, SmAP (S. mansoni alkaline phosphatase) can cleave AMP to generate adenosine.

To confirm that SmATPDase1 is a true apyrase, i.e., it can cleave ATP and ADP to yield AMP and Pi, as suggested by the gene knockdown experiments, a recombinant form of the protein was expressed in CHO-S cells. Attempts were made to generate a soluble form of SmATPDase1 (lacking transmembrane domains but retaining all key enzymatic motifs). While this goal was achieved, the soluble protein was enzymatically inactive, likely due to incorrect post-translational handling. In contrast, a full-length version of SmATPDase1 was generated in CHO-S cells that was active. The retention of the two terminal, transmembrane domains in this full-length recombinant protein seems important for proper folding and may help to maintain the protein in an enzymatically favorable conformation. There is firm evidence from work with ATPDases in other systems that the interaction of the transmembrane domains and their mobility in a lipid bilayer regulate enzyme catalysis (Knowles, 2011). For instance, the extracellular domain of the chicken NTPDase8 enzyme has a small fraction of the activity of the full length enzyme (Li, Lee & Knowles, 2010). At least some of the full length SmATPDase1 ectoenzyme expressed here is found in the plasma membrane of the CHO-S cells such that live intact cells expressing the protein display both ATP and ADP cleaving capabilities.

The ability of recombinant SmATPDase1 expressed in CHO-S cells to hydrolyze ATP and ADP was found to be enhanced by the addition of divalent cations to the mixture; adding Ca++ or Mg++ to the rSmATPDase1 preparation greatly increases activity. Adding Mg++ or Ca++ to schistosome tegument preparations had previously been shown to promote ATPase and ADPase activity (Torres et al., 1998; Vasconcelos et al., 1993). As reported here, removing these cations from the rSmATPDase1 preparation by the addition of the chelating agents EDTA plus EGTA effectively shuts down the enzyme. The fact that SmATPDase1 is a calcium-activated plasma membrane-bound enzyme again confirms it as a member of the apyrase family. Earlier substrate competition experiments (ATP versus ADP) involving schistosome tegument extracts, as well as comparative heat inactivation profiles for ATP versus ADP hydrolytic activities using these extracts, led to the hypothesis that a single enzyme in the tegument was responsible for degrading both ATP and ADP (Martins, Torres & Ferreira, 2000; Vasconcelos et al., 1993). Our work confirms this hypothesis. Furthermore, under physiological conditions the Km of rSmATPDase1 for ATP is 0.4 ± 0.02 mM and for ADP is 0.252 ± 0.02 mM and these values are almost identical to those reported for the ATPase activity and the ADPase activity of adult S. mansoni tegumental extracts (0.25 mM for ADP and 0.45 mM for ATP, Vasconcelos et al., 1993).

As discussed, the apyrase activity detected in schistosome tegument membrane preparations has a very similar profile to that described here for rSmATPDase1. One difference is apparent, however; the ATPase activity of the tegument preparation (but not its ADPase activity) has been reported to be inhibited by thapsigargin in a dose dependent manner (Martins, Torres & Ferreira, 2000). Inhibition of ∼70% was seen with 100 µM thapsigargin (Martins, Torres & Ferreira, 2000). This finding was a surprise since thapsigargin is best known as a specific inhibitor of sarco/endoplasmic reticulum Ca++ (SERCA) ATPases and not of apyrases (Rogers et al., 1995). Our finding is that there is no dose-dependent inhibitory effect of thapsigargin on rSmATPDase1 activity; at all thapsigargin concentrations tested (10–100 µM) ATPase activity is decreased by ∼20% and ADPase activity by ∼5%. Our data show that, as for other apyrases, the SmATPDase1 enzyme is not intrinsically inhibitable by thapsigargin in a dose dependent manner. The inhibition reported (Martins, Torres & Ferreira, 2000) is likely related to the use of tegument preparations rather than recombinant enzyme.

As shown here, both ATP and ADP cleavage activities of rSmATPDase1 are more pronounced in an alkaline environment. Earlier, the ATPase activity detected in adult schistosome tegumental membrane extracts was reported to be similarly enhanced under alkaline conditions (Cesari, Simpson & Evans, 1981). Furthermore, other enzyme activities (alkaline phosphatase and phosphodiesterase) detectable in tegument extracts are likewise greatest at pH > 9 (Cesari, Simpson & Evans, 1981). Why the three ectoenzymes SmAP, SmNPP-5 and SmATPDase1, expressed at the host-parasite interface, should all display highest activity under alkaline conditions is unclear. Perhaps schistosomes in vivo maintain an alkaline environment immediately around them in which these enzymes optimally act and which has some selective advantage for the worms.

Our work demonstrates that the three ecto-enzyme genes SmAP, SmNPP-5 and SmATPDase1 can all be specifically and strongly knocked down using target specific siRNAs. It is noteworthy that roughly equivalent suppression is obtained irrespective of whether each gene is targeted alone or together with the other two targets. In other words there is no compromise in suppression efficiency when all three genes are targeted together, demonstrating that the RNAi machinery in schistosomes is not saturated by multiple siRNAs targeting different mRNAs at the same time.

Just as important for schistosomes as the elimination of potentially damaging host signaling molecules like ATP and ADP by SmATPDase1 may be the generation of adenosine from AMP via SmAP. This is because many of the proinflammatory effects of ATP on immune cells can be suppressed or reversed by adenosine (Reviewed in Bours et al., 2006; Hasko & Cronstein, 2004). Extracellular adenosine can impede the chemotactic responses of macrophages and monocytes and can inhibit both their production of pro-inflammatory cytokines as well as macrophage proliferation, phagocytosis and lysosymal enzyme secretion (Bours et al., 2006; Riches et al., 1985). Extracellular adenosine can inhibit the production of reactive nitrogen species and reactive oxygen species by monocytes/macrophages and neutrophils (Bours et al., 2006; Flamand et al., 2000). In addition, adenosine can impede lymphocyte adhesion and attenuate the proliferative and cytotoxic responses of activated T cells (Bours et al., 2006; Hasko & Cronstein, 2004).

On a more mundane level, the adenosine generated by this pathway may be directly taken up by schistosomes as food (Levy & Read, 1975b). The ATP catabolic pathway may be used to generate purine derivatives in the vicinity of the worms that can then be easily imported and this function may have particular importance for schistosomes since the parasites are unable to synthesize purines de novo (Levy & Read, 1975a). The hypotheses that any adenosine generated via this pathway may be taken in by the parasites as food or may act to impede host purinergic signaling are not mutually exclusive.

Since the 3 ecto-enzymes may make good vaccine candidates, all have been purified from inclusion bodies following their expression as recombinant proteins in E. coli (Rofatto et al., 2013). In vaccine trials, immunization with the isolated individual proteins, or with all three proteins combined, did not reduce the worm burden of challenged mice. However, immunization with SmAP alone or with all three proteins together, when combined with subcurative treatment with the drug praziquantel, was able to reduce worm burdens by ∼40% (Rofatto et al., 2013).

An ability to cleave ATP and/or ADP in the extracellular environment has been described in several pathogens. For example ecto-ATPase activity has been described in the protozoan parasites Toxoplasma gondii (Bermudes et al., 1994), Leishmania amazonensis (Berredo-Pinho et al., 2001), Trichomonas vaginalis (de Jesus et al., 2002) and Cryptosporidum parvum (Manque et al., 2012) and in several bacterial pathogens including Mycobacterium bovis (Zaborina et al., 1999), Vibrio cholera (Punj et al., 2000), Staphylococcus aureus (Thammavongsa et al., 2009), and Legionella pneumophilia (Vivian et al., 2010). Similarly, blood-feeding ectoparasites are known to release a repertoire of nucleotide-metabolizing enzymes in their saliva (Andersen et al., 2007; de Araujo et al., 2012). The parasitic nematode Trichinella spiralis secretes a panel of nucleotide metabolizing enzymes (Gounaris, 2002). Thus a conserved feature of several pathogens, including schistosomes, is an ability to control local ATP and ADP levels, perhaps to thereby inhibit inflammation and thrombosis and protect the pathogens. Identifying chemical inhibitors of SmATPDase1 to negate the worm’s ability to degrade exogenous pro-inflammatory and pro-thrombotic nucleotides may offer a novel therapeutic option to treat schistosomiasis.

Infected snails were provided by the Biomedical Research Institute via the NIAID schistosomiasis resource center under NIH-NIAID Contract No. HHSN272201000005I. We thank Dr. Chuck Shoemaker for helpful discussion and Dr. Sergio Verjovski-Almeida, University of Sao Paulo, Brazil for the anti-SmATPDase1 antibody.

Additional Information and Declarations

Competing Interests

Author Contributions

Animal Ethics

The authors declare they have no competing interests.

Akram A. Da’dara conceived and designed the experiments, performed the experiments, analyzed the data, wrote the paper, prepared figures and/or tables, reviewed drafts of the paper.

Rita Bhardwaj conceived and designed the experiments, performed the experiments, analyzed the data, prepared figures and/or tables, reviewed drafts of the paper.

Yasser B.M. Ali performed the experiments, reviewed drafts of the paper.

Patrick J. Skelly conceived and designed the experiments, analyzed the data, wrote the paper, prepared figures and/or tables, reviewed drafts of the paper.

The following information was supplied relating to ethical approvals (i.e., approving body and any reference numbers):

Tufts University IACUC Protocol#: G2012-150.

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
