# Peer review of "Schistosome tegumental ecto-apyrase (SmATPDase1) degrades exogenous pro-inflammatory and pro-thrombotic nucleotides"

_PeerJ, doi:10.7717/peerj.316_

## Round 0.1 · original submission · Minor Revisions

1. Please indicate how many replicates were used to generate each mean value depicted in all Figures. Also define error bars as indicating SEM or SD. In addition, it would also be useful to know how many times experiments were repeated.

2. In Figure 1, please show background (i.e. minus parasite) hydrolysis rates to demonstrate that hydrolysis is due to worm enzyme activity.

·

Basic reporting

This is a clearly written and insightful manuscript on a fascinating and potentially important aspect of schistosome biology. The introduction and discussion are thorough and appropriately reference prior literature.

Experimental design

The authors are to be commended for examining nucleosidase function using RNAi - the most rigorous and direct approach currently available for this kind of gene function study in schistosomes. This is challenging and difficult work.

Validity of the findings

The experiments are beautifully executed and the data are robust and convincing. I had only a couple of minor concerns regarding the presentation and interpretation of the data:

Figure 1 – would have been helpful to include inorganic phosphate accumulation for control wells lacking parasites, to get a sense of the background levels of hydrolysis in this assay, if any.

Paragraph starting line 284 – RNAi does appear to specifically reduce the levels of protein present in the treated parasites. However, I would avoid use of descriptors such as “substantial” (line 288) unless the reduction has been actually quantified (by densitometry etc).

It is puzzling that the SmATPDase displays optimal activity for ATP at a slightly alkaline pH. One would have expected that activity would be maximal at neutral pH. Could this be an adaptation to exhibit maximal activity when under attack from immune cells, where an oxidative burst would presumably be associated with local alkalization? Is there any evidence that expression of SmATPDase or any of the other nucleosidases is up-regulated as part of a stress response to external insults such as oxidative stress?

Reviewer 2 ·

Basic reporting

The work is well posited, properly presented and well documented including updated references. The conclusions are in accordance with the results obtained.

Experimental design

The methodology used is appropriate and justified, and the different techniques employed are in accordance with the proposed aims. In general, this methodology is described in a clear, rigorous and detailed way.

Validity of the findings

The results of the experiments are clear. However, in order to evaluate their soundness, authors should clearly indicate how many replications within each group are performed in the experiments. Equally it should be indicated on the Figures if the represented values are the mean of various essays and, if this is the case, indicate the number as well. It is not specified to that correspond the error bars, if they represent standard deviation or standard error.
Erroneously the table 1 has been titled as Figure 7.

Additional comments

The work of Da'dara et al. is an interesting article that provides novel and relevant information in the field. Once corrected or clarified the indicated points, I consider that the work can be published in PeerJ.

---

## Round 0.2 · Minor Revisions

Dear Prof Skelly,
Thank you for your revised manuscript, and my apologies for taking a few days to review it. I feel that your revisions mostly address the criticisms raised by the reviewers, and that the work is suitable for PerrJ. However, the meaning of the repeated statement "The data shown are representative of at least x independent experiments" is ambiguous, having two possible readings - 1 or 2 below:

(1) The statement could mean that for each histogram shown for an experiment, n≥x, and the data points contributing to each mean were generated in separate experiments, where n=1 in each repeat. In other words, the data are not "representative of ... x experiments", but are, in fact, a summary of all data collected, and that n≥x.

(2) The statement could mean that the experiment was performed ≥x times with a number of replicates (n) each time, and that the data shown is from one of those experiments. (This is what I take "data are representative of x experiments" to usually mean.) In this case, the number of replicates "n" for the actual representative experiment shown, has still not been given, but should be given.

Could you re-word the figure legends to ensure that only one, and not both, of the above two readings are implied please?

regards
Chris Jolly

---

## Round 0.3 · accepted · Accept

Dear Prof Skelly. Thanks for the revisions and for your rebuttal email. The paper reads very clearly now, and I think makes a worthy contribution to PeerJ.